# A Straightforward Approach to Synthesize 7-Aminocephalosporanic Acid In Vivo in the Cephalosporin C Producer *Acremonium chrysogenum*

**DOI:** 10.3390/jof8050450

**Published:** 2022-04-26

**Authors:** Xuemei Lin, Jan Lambertz, Tim A. Dahlmann, Marc M. Nowaczyk, Burghard König, Ulrich Kück

**Affiliations:** 1Allgemeine und Molekulare Botanik, Ruhr-Universität Bochum, 44780 Bochum, Germany; xuemei.lin@rub.de (X.L.); tim.dahlmann@rub.de (T.A.D.); 2Plant Biochemistry, Ruhr-University Bochum, 44780 Bochum, Germany; jan.lambertz@rub.de (J.L.); marc.m.nowaczyk@rub.de (M.M.N.); 3Koenig und Funk Biotech GmbH, 13156 Berlin, Germany; bk@kfbiotech.de

**Keywords:** 7-Aminocephalosporanic acid, cephalosporin C acylase, *Acremonium chrysogenum*

## Abstract

The pharmaceutical industry has developed various highly effective semi-synthetic cephalosporins, which are generated by modifying the side chains of the core molecule 7-aminocephalosporanic acid (7-ACA). In industrial productions, the 7-ACA nucleus is obtained in vitro from cephalosporin C (CPC) by chemical or enzymatic processes, which are waste intensive and associated with high production costs. Here, we used a transgenic in vivo approach to express bacterial genes for cephalosporin C acylase (CCA) in the CPC producer *Acremonium chrysogenum*. Western blot and mass spectrometry analyses verified that the heterologous enzymes are processed into α- and β-subunits in the fungal cell. Extensive HPLC analysis detected substrates and products of CCAs in both fungal mycelia and culture supernatants, with the highest amount of 7-ACA found in the latter. Using different incubation times, temperatures, and pH values, we explored the optimal conditions for the active bacterial acylase to convert CPC into 7-ACA in the culture supernatant. We calculated that the best transgenic fungal strains exhibit a one-step conversion rate of the bacterial acylase of 30%. Our findings can be considered a remarkable contribution to supporting future pharmaceutical manufacturing processes with reduced production costs.

## 1. Introduction

Cephalosporin antibiotics are a major group of beta-lactam antibiotics derived from the filamentous fungus *Acremonium chrysogenum*. They are important for hospital patients in preventing and treating infectious diseases such as bone, ear, skin, urinary, and upper respiratory tract infections. Cephalosporins inhibit bacterial growth by disrupting the synthesis of the peptidoglycan layer in bacterial cell walls. Cephalosporin C (CPC) was the first cephalosporin antibiotic compound isolated, and its chemical structure was characterized soon after its discovery by Guiseppe Brotzu in 1945 [1,2]. CPC and its derivatives have broad antibacterial activity against Gram-positive and Gram-negative bacteria.

Cephalosporins are categorized into five generations, depending on their spectrum of antibacterial activity and their sequential order of discovery. The first-generation cephalosporins are active against most Gram-positive cocci, including methicillin-susceptible *Staphylococcus aureus* (MSSA) and streptococci, but have weak activity against Gram-negative bacteria [3]. This disadvantage was circumvented when the second- and third-generation cephalosporins reached the pharmaceutical market. These cephalosporins possess higher stability against beta-lactamases from Gram-negative bacteria such as *Haemophilus influenzae* [3]. Another advantage of second-generation cephalosporins is that one can reduce dosage and extend administration intervals during the treatment of infections [4]. Also, second- and third-generation cephalosporins avoid the risk of cross-reactivity in penicillin-allergic patients since the cross-reaction caused by the similar chemical structures at the side chains of first-generation cephalosporins and penicillins no longer applies [5].

Compared with third-generation cephalosporins, the fourth-generation of cephalosporins have broader activity against Gram-negative bacteria due to rapid penetration through the outer membrane, and also lower affinity to beta-lactamases, which resulted in resistance to previous generations [6]. Additionally, some fourth-generation cephalosporins have excellent efficacy against Gram-positive bacteria such as MSSA, penicillin-resistant pneumococci, and some streptococci [6]. These benefits have led to the change to using fourth-generation cephalosporins to treat serious infections in hospitalized patients [7].

The fifth-generation cephalosporins are also broad-spectrum antibiotics, uniquely including activity against methicillin-resistant *Staphylococcus aureus* (MRSA) [8]. MRSA’s resistance mechanism involves producing mutant antibiotic targets, which inhibits the antibacterial activity of cephalosporins by reducing the binding affinity to the beta-lactam rings [9].

A common feature of all cephalosporins is that they are chemically derived from a starting compound, either 7-aminocephalosporanic acid (7-ACA) or 7-amino-deacetoxycephalosporanic acid (7-ADCA), the core building blocks of semi-synthetic cephalosporin antibiotics. Initially, 7-ACA was produced from CPC through a series of chemical reactions [10,11]. However, the chemical production method resulted in severe environmental impacts, such as the requirement for organic solvents and the release of toxic chemical waste and polluted water [11,12]. Since two-step enzymatic procedures have a yield similar to chemical production, this gradually replaced the classical chemical route to producing 7-ACA. For the first step, D-amino acid oxidase (DAO), produced by *Trigonopsis variabilis*, is used as an immobilized biocatalyst for modifying the 7-aminoadipyl side chain of CPC, generating glutaryl-7-aminocephalosporanic acid (GL-7-ACA). For the second step, GL-7-ACA acylase (GLA), a heterologous recombinant protein produced and isolated from *Escherichia*
*coli,* is used as an immobilized enzyme for further conversion of GL-7-ACA to the desired product, 7-ACA [13]. This two-step enzymatic method is economically sustainable and efficient due to the vastly reduced amount of waste and the mild reaction conditions required in terms of temperature and pH [11,14,15].

The latest innovation in 7-ACA production is a one-step bioconversion using a single enzyme, namely cephalosporin C acylase (CCA). CCAs, discovered and developed from various bacterial strains, have been classified into five classes based on their gene structures, molecular masses, and enzyme properties [16,17]. Over the last two decades, numerous protein engineering procedures have been conducted on potential CCAs, such as bacterial acylases and glutaryl amidases, to obtain mutant variants cleaving the acyclic amide bond in CPC [17,18,19,20,21]. Substrate specificity was improved using molecular modeling and site-directed mutagenesis approaches. For example, six amino acids of the GL-7-ACA acylase from *Pseudomonas* sp. *SE83* were modified, and the corresponding mutant was called S12. Compared to wild-type, this S12 showed an 850% increase in activity on the substrate CPC [22].

Pharmaceutical companies have used CCAs with high activities since 2006 [15]. These one-step conversion CCAs are produced and isolated from *E. coli* and immobilized in vitro on a solid support material to stabilize as a biocatalyst [23]. Until now, the successful industrial application of CCA in vivo has not been reported, although the simultaneous biosynthesis of CPC and bioconversion from CPC to 7-ACA in one pot, i.e., within the CPC producing organism, would avoid CPC isolation steps and enzymatic operations. By minimizing sophisticated downstream processes, this straightforward approach to producing 7-ACA in vivo would greatly contribute to economically and ecologically sustainable antibiotic manufacture.

*Acremonium chrysogenum* is the exclusive natural producer of CPC in the pharmaceutical industry. So far, reports about recombinant *A. chrysogenum* strains are limited due to the peculiar growth characteristics of production strains, namely their slow growth rate and lack of conidiospores. Furthermore, the chemical instability of CPC compared to penicillin has limited the generation of stable production lines by preventing easy propagation and genetic engineering.

Previously, we and others have developed a broad range of molecular tools for genetically engineering *A. chrysogenum* [24,25]. Here, we describe the construction of recombinant fungal strains producing a high yield of recombinant bacterial CCAs, followed by a one-step conversion of CPC to 7-ACA.

## 2. Materials and Methods

### 2.1. Strains, Plasmids, and Culture Conditions

Recombinant plasmids were constructed using either standard laboratory techniques or the Golden Gate Assembly method with *Escherichia coli* strain XL1-Blue MRF’ as the host for plasmid amplification and maintenance [26,27,28].

*A. chrysogenum* strains used in this study are listed in Table 1. Fungal strains were grown and maintained on solid complete culture media (CCM) at 27 °C with full-time light exposure [29]. Liquid fungal cultures were incubated in Erlenmeyer flasks in 100 mL CCM or minimal medium (MM) at 27 °C with 180 rpm shaking speed [30]. As a preculture, a small amount of *A. chrysogenum* mycelia were inoculated into 100 mL of liquid CCM medium. After 7 days of incubation, the arthrospores were harvested by centrifugation (5000× *g*) for 5 min, and the resulting arthrospore pellet was dissolved to a density of 200 mg/mL with 0.9% NaCl solution. For every experiment, 200 mg of arthrospores were inoculated into liquid CCM to ensure equal starting cell weights. MM was used for all protein isolation processes involved in experiments. DNA-mediated transformation of *A. chrysogenum* was performed with protoplasts as described previously [29,31]. The resulting transformants were selected on CCM solid medium containing nourseothricin at a concentration of 25 µg/mL [32].

### 2.2. Tools for Codon Adaptation, Gene Synthesis, and Protein Analysis

Codon-optimization processes were performed for three bacterial *cca* genes using GENEius software (Eurofins Genomics; Ebersberg, Germany). The codon usage of *A. chrysogenum* referred to the codon usage database (https://www.kazusa.or.jp/codon/cgi-bin/showcodon.cgi?species=5044, accessed on 18 October 2018) from the Kazusa DNA Research Institute (Kisarazu, Japan). Designed genes were custom synthesized by Eurofins Genomics (Ebersberg, Germany). Signal peptide prediction was performed by SignalP-5.0 (https://services.healthtech.dtu.dk/service.php?SignalP-5.0, accessed on 8 May 2020).

### 2.3. Construction of Fungal Gene Expression Vectors Using Codon-Adapted Bacterial Acylase Genes

To construct the fungal gene expression vector pAB-nat, oligonucleotides MCS_pAB and MCS_pAB-rev (Appendix A) were annealed, digested with *Not*I and *Bam*HI, and ligated into *Not*I and *Bam*HI-digested plasmid pGG-C-EGFP (Table 2), thereby replacing the *egfp* and *lacZ* genes and inserting two new *Bsa*I sites for further Golden Gate cloning. For selecting fungal transformants, the vector molecule carries the nourseothricin resistance gene, allowing growth on antibiotic-containing media [33,34].

The codon-optimized *ccaA*, *ccaB*, and *ccaC* genes were fused with Hisx6- or HA-tags at 3′, and HA-tags at 5′ termini, *Bsa*I restriction enzyme recognition sites were added at both ends. The custom synthesized *ccaA* gene cassette was integrated into the plasmid pEX K248 >CCAA, the *ccaB* gene into pEX-K248->CCAB, and the *ccaC* gene into pEX-K168->CCAC. All plasmids carry the kanamycin resistance gene as a selection marker.

To construct the designed fungal expression vector pXUL-4, the *ccaA* gene was amplified from pEX K248 >CCAA by Phusion High-Fidelity DNA polymerase PCR (Thermo Scientific; Dreieich, Germany) using the primer pair of His-CCAA-ATGdel_F and CCAA_R; both primers include the *Bsa*I restriction enzyme recognition sites at their 5′ terminus. The amplified PCR fragment was inserted into the backbone of the fungal expression vector pAB-nat using Golden Gate Assembly [37]. The fungal plasmid for the expression of recombinant *ccaB* (pXUL-22) was generated by Golden Gate Assembly of plasmid pEX-K248->CCAB and pAB-nat. To express the recombinant *ccaC*, fungal expression vector pXUL-2 was generated with the same method using the plasmid pEX-K168->CCAC and pAB-nat. pXUL-10 was generated to investigate the subcellular localization of CCA; this plasmid contains the EGFP coding gene tagged at the 3′ end of the recombinant *ccaA*. The primers containing *Bsa*I restriction sites (pXUL-10_F-2, pXUL-10_R-2) were used to amplify the His-*ccaA*-HA cassette from pXUL-4, and the amplified fragment was inserted into pGG-C-EGFP vector by Golden Gate cloning.

### 2.4. Protein Extraction and Western Blot Analysis

For crude protein extractions, *A. chrysogenum* strains were grown in liquid MM for 24–144 h, as described above. The protein extraction procedure followed a conventional protein extraction method for filamentous fungi [38]. The protein concentration was determined for each sample using the Bradford assay [39]. Using 15 µg of protein from each fungal transformant, SDS-PAGE was performed with a 12% separation gel and 5% stacking gel, followed by standard procedure Western blot analysis. Anti-HA antibody (Sigma Aldrich; Burlington, VT, USA) and HRP-linked anti-mouse IgG antibody (Cell Signaling Technology; Frankfurt am Main, Germany) were used as the primary and secondary antibodies. The SuperSignal™ West Pico PLUS Chemiluminescent Substrate (Thermo Scientific; Dreieich, Germany) was used as the signal detection reagent, and imaging was performed by the Chemidoc XRS+ System (Bio-Rad; Hercules, CA, USA) with an exposure time of 5 s.

### 2.5. Fluorescence Microscopy

Microscopic investigations were conducted using an Axio Imager M1 microscope (Zeiss; Oberkochen, Germany) as described recently [35,40]. Recombinant strains expressing EGFP-CCAA, were grown in liquid CCM medium for 72 h. Microscopic detection was performed on microscope slides with 10 µL of vegetative mycelia diluted with the same volume of 0.9% NaCl solution. Nuclei and mitochondria were stained with DAPI in a 5 mg/mL concentration diluted in 0.9% NaCl solution. Images were acquired and analyzed using Metamorph software (Version 7.7.0.0; Universal Imaging, Bedford Hills, NY, USA).

### 2.6. Southern Blots

Genomic DNA extraction from *A. chrysogenum* and Southern blot analysis were performed as previously described [41]. Copy numbers were quantified using ImageJ software [42]. The amount of genomic DNA loaded for each sample was normalized to the hybridization signal of the reference gene encoding α-tubulin. Tandem integration of gene copies into the fungal genome leads to highly accumulated hybridization signals at some locations, as visible on the autoradiogram.

### 2.7. HPLC Analysis of Acylase Substrates and Products from Mycelia and Supernatants

Mycelial extracts were measured with cells from 72 h of culture. The mycelia were collected and homogenized with liquid nitrogen. Extraction was performed on 250 mg of ground mycelia in 1 mL of 50 mM Na_2_HPO_4_ (pH 6.0 adjusted with H_3_PO_4_). Considering the wet weight of mycelia after 72 h of growth is around 0.02 g/mL, the mycelial extract used for HPLC analysis was concentrated 12.5 times. *A. chrysogenum* culture supernatants were harvested after 24–168 h of culture by vacuum filtration through Whatman^®^ filter paper. To remove protein residues from fungal mycelial extracts and culture supernatants, the same volume of methanol was added, followed by vigorous vortexing and overnight freezing, and the precipitated proteins were eliminated by centrifugation at 16,000× *g* rcf for 10 min, followed by membrane filtration with a 96-well filter (AcroPrep Mustang S, Pall Corporation; New York, NY, USA).

For every HPLC measurement with the Agilent 1100 HPLC (Santa Clara, CA, USA), 2 µL of the sample was injected into a separation column (Agilent ZORBAX Eclipse Plus C18 (Santa Clara, CA, USA); 4.6 × 150 mm, 5 µm). HPLC-grade water (Eluent A) and 90% acetonitrile (ACN) (Eluent B), both containing 0.5% phosphoric acid, were used as the mobile phase eluents, using a gradient elution (Appendix A) designed for optimal compound elution with a flow rate of 1.2 mL/min. The Agilent 1100 Series G1315B Diode Array Detector (Santa Clara, CA, USA) was used as a UV detector measuring wavelengths of 210 nm and 260 nm. The retention time of major compounds was defined by measuring standard substances dissolved in 50 mM Na_2_HPO_4_ solution using the same measurement method as the samples. As standard compounds, we used 7-ACA, 7-ADCA, hydroxymethyl-7-amino-cephalosporanic acid (HACA), deacetoxycephalosporin C (DAOC), and CPC provided by Sandoz AG of Novartis (Basel, Switzerland).

### 2.8. Protein LC-MS/MS Analysis of Fragmented Peptides from CCAA and CCAB

As a protein separation step, SDS-PAGE was performed with the protein samples extracted from the mycelia from *ccaA* and *ccaB* transformants of 6 days of growth in liquid MM. Growth of strains and protein extraction was performed as described above. SDS-PAGE gels were stained with Coomassie staining solution (0.2% (*w*/*v*) Coomassie Blue, 7% (*v*/*v*) acetic acid, 50% (*v*/*v*) methanol), and the protein bands of 79 kDa in size (precursor), 60 kDa (β-subunit) and 19 kDa (α-subunit) were cut out and prepared for mass spectrometry (MS) analysis as described previously [43].

In brief, the gel pieces were washed in 50 mM ammonium bicarbonate (ABC) and destained at least 3 times for 20 min in 25 mM ABC in 50% (*v/v*) ACN. The gel pieces were dried in 100% (*v/v*) ACN and treated overnight with MS-grade trypsin or chymotrypsin (Promega, Madison, WI, USA) in 25 mM ABC. The peptides were extracted twice from the gel pieces with 1% (*v/v*) formic acid, 50% (*v/v*) ACN, combined with the supernatant of the digest, dried in a vacuum concentrator, and stored at −20 °C until use. Before the measurement, the peptides were resuspended in solution A (1% (*v/v*) formic acid, 2% (*v/v*) ACN in water). LC-MS/MS analysis was performed with a previously described setup consisting of a nanoACQUITY gradient UPLC (Waters; Milford, CT, USA) and an Orbitrap ELITE mass spectrometer (Thermo Fisher Scientific; Waltham, MA, USA) using a discontinuous gradient over 60 or 180 min from 2–85% (*v/v*) ACN [43,44]. Non- and single-charged ions were excluded from the measurement.

Data processing and analysis were done using ProteomeDiscoverer Version 2.3 (Thermo Fisher Scientific; Waltham, MA, USA), with the proteome database from *A. chrysogenum,* which was extended by the introduced proteins, N-terminal variations, and common contaminants [45]. The false-discovery rate (FDR) was set to 1%, and the following modifications were allowed: M-oxidation, NT-acetylation. A full summary of the data processing can be found in the Appendix A.

## 3. Results

Here, we investigated the in vivo synthesis of three different CCAs in *A. chrysogenum*. The enzymes (CCAA, CCAB, and CCAC) are distinguished by their physico-chemical properties, and their main features are given in Table 3. For example, CCAA and CCAB comprise 98% identical amino acid sequences, and both were derived from a class I CCA isolated from *Pseudomonas* sp. GK16 [46,47]. CCAC is a mutant variant of AcyII, an acylase that was identified from *Pseudomonas* sp. SE83 and classified as class III [48]. Despite belonging to different classes of CCAs, they share the α-subunit/spacer/β-subunit arrangement and a conserved sequence at crucial catalytic regions [19]. Previous studies demonstrated that some mutant variants of CCAs show improved substrate activities towards CPC in vitro. For example, the deacylation activity of CCAB on CPC was increased by 25.3 times, and CCAC activity was 9.6 times higher than the wild-type acylase [22,49,50].

### 3.1. Gene Synthesis and Vector Construction for Introducing Three Codon-Optimized Cca Genes into A. chrysogenum

The expression of heterologous genes in filamentous fungi was previously optimized by using synthetic genes with a codon-optimized open reading frame [34,52]. For the efficient expression of the three bacterial acylase genes in *A. chrysogenum,* we designed codon-optimized genes *ccaA*, *ccaB*, and *ccaC* to ensure adequate tRNA supply during the protein translation process in the fungal host (Figure 1). The frequency of each codon used in the optimized *ccaC* gene is close to that of the *A. chrysogenum* genome. The mean difference in the entire codon usage between *A. chrysogenum* and the bacterial *ccaC* gene was considerably reduced from 21.09% to 5.53% (Figure 1). Simultaneously, the GC content of the codon-optimized genes was adjusted to around 61% to adapt to the high GC content of coding sequences found in *A. chrysogenum* [45]. Codons were similarly optimized for the *ccaA* and *ccaB* genes (Appendix A).

### 3.2. Bacterial Acylase Expression in A. chrysogenum

To heterologously express *cca* genes in *A. chrysogenum,* we used vectors that carry *cca* genes under the control of the constitutive *gpdA* promoter [53] (Figure 2). Randomly selected nourseothricin-resistant transformants were investigated further at the protein level. Total protein extracts were isolated from the mycelia of CCA transformants and analyzed by Western blotting. To detect the recombinant enzymes, we used an anti-HA tag antibody, which detected a significant amount of the recombinant CCAs in transformants carrying the three recombinant enzymes (Figure 3A–C).

The variable amounts of protein in randomly selected transformants may be explained by the copy number of the recombinant genes. Transformation using plasmid DNA results in ectopic integration into genomic DNA of the fungal host [54]. This was confirmed by Southern blot hybridization experiments (Appendix A). The gene encoding α-tubulin of *A. chrysogenum* was selected as an internal standard to normalize the amount of genomic DNA loaded on the agarose gel. The host strain for transformation (XUL-pAB-nat) served as a control. From the signal intensity on the Southern blot autoradiogram, we calculated 17, 4, and 11 copies of the *ccaA* gene in the selected transformants. In subsequent experiments, we used the recombinant fungal strains showing the best bacterial enzyme synthesis.

Next, we addressed the question of where the acylase protein is located within the fungal cell. For fluorescence microscopy, we constructed a CCA-GFP expression vector, and after transformation, recombinant strains were screened for GPF fluorescence. The presence of the full-length CCA-GFP protein was confirmed by Western blotting (Appendix A). Fluorescence was detected within the cytoplasm compared to the control nuclei stained with DAPI (Figure 4). We further surmise that the vacuoles contain only a minor amount of the recombinant enzyme.

For enzymatic activation, autocatalytic cleavage of the CCA enzyme is necessary, where the precursor polypeptide is cut into an α-subunit and β-subunit through a series of hydrolytic reactions led by the Ser1β amino acid [19,55,56,57]. For CCAA and CCAB transformants, we succeeded in detecting proteins bands resembling the β-subunit (60 kDa), which emerged after cleavage. The precursor polypeptide bands (79 kDa) were also detected, but with lower signal intensity (Figure 3A,B). These results verify that the bacterial *cca* genes were successfully expressed in *A. chrysogenum*. Only CCAC was unable to undergo autocatalytic cleavage in the fungal cell and was thus excluded from further analysis (Figure 3C). In all our Western blot analyses, we failed to detect the α-subunit from CCAA and CCAB.

### 3.3. Evidence for Alpha and Beta CCA Subunits by MS Analysis

Since the Western blot data above showed no evidence of a processed α-subunit in any of the transformants, we chose an alternative way to detect the proteins and performed an extensive protein mass spectrometry analysis (LC-MS/MS) on transformants expressing either CCAA or CCAB. Due to their 98% sequence similarity, the data sets were combined (Figure 5). Protein extracts were separated by polyacrylamide gels, and gel slices for the enzyme precursor, α-subunit, and β-subunit were selectively cut out and digested with trypsin or chymotrypsin, respectively (Appendix A).

Protein MS detected several peptides from all parts of the enzyme (Figure 5; Appendix A). Due to the high sensitivity of protein MS analysis, we were able to detect peptides covering around 48% of the CCA in transformants producing either CCAA or CCAB. The most unexpected finding was that the first peptide from the N-terminus of the α-subunit, generated from trypsin digestion, was truncated (Appendix A). The detected peptide sequence was STPQAPIAAYKPR, instead of the expected sequence MHHHHHHHGGGGSEPTSTPQAPIAAYKPR. Apparently, the HisX6-tag, the GS-linker, and three amino acids (EPT) of the α-subunit are removed after protein translation. This N-terminal amino acid truncation may explain why we were unable to detect the α-subunit in Western blots. However, most of the remaining parts of the α-subunit were detected. These include the complete peptides at the C-terminus of the α-subunit, confirming the accurate cleavage between the α-subunit and spacer (Appendix A). Many peptides representing the β-subunit were also confirmed to be correctly synthesized. Remarkably, the final peptide generated by the trypsin digestion contained the full sequence of the HA-tag. However, the low number of peptide sequence matches (PSMs) of CCA-peptides compared to other proteins identified in the gel slices may indicate a low abundance of the CCA protein.

### 3.4. Time-Dependent Processing of the CCA Precursor and Detection in Mycelia and Supernatants

Next, we asked how acylase precursors are processed in mycelial protein extracts. We followed the time course of the expression and cleavage process of CCAA and CCAB from mycelia harvested after up to 144 h in culture (Figure 6A,B). Two strains expressing the *ccaA* or *ccaB* genes (XUL-4.6 and XUL-22.3) were selected for further investigations due to their high CCAA and CCAB production levels (Figure 3A,B). The cleavage process continued gradually and was completed by the end of the chosen culture time (144 h). The amount of CCAA β-subunit was maintained at a high level, which is potentially a part of the active CCAA enzyme in fungal culture (Figure 6A). In contrast, CCAB production was slower than that of CCAA. The strongest protein precursor signal was observed after 144 h of culture (Figure 6B). The cleavage process was also slower. The highest amount of the β-subunit was detected after 144 h and represented approximately half the amount of the enzyme precursor.

Interestingly, a high amount of β-subunit was also detected in protein extracts of concentrated culture supernatants (Figure 6C). The signal for the β-subunit of CCAA (XUL-4.6) was significantly stronger than that of CCAB (XUL-22.3), and the precursors of both were also observed as faint bands. According to predictions about signal peptides, CCAA and CCAB do not contain a known fungal or bacterial signal peptide guiding the secretion of protein to the extracellular space. Therefore, they might be exported out of the fungal cell by so far unknown or unspecific protein transporters. In essence, Western blot analysis shows the presence of the β-subunit in culture supernatants and predicts that an active CCA enzyme that converts CPC to 7-ACA in *A. chrysogenum* cultures is possible.

### 3.5. Detection of Substrates and Products of CCAs in Fungal Mycelia and Culture Supernatants

Since we detected a decent amount of CCAs in the culture supernatants of *A. chrysogenum*, we wanted to discover where the intermediate and end products of the CPC biosynthesis pathway are located in general. HPLC analyses were performed to investigate the intracellular and extracellular distribution of such substrates and products. Interestingly, completely different spectra were generated after HPLC analysis of mycelia and supernatants, whereas differences between wild-type and fungal transformants were not apparent (compare Figure 7A,B). Using standard compounds, we identified peaks of the intermediate product DAOC, the end product CPC, 7-ADCA (the 7-deacylation product of DAOC), and 7-ACA (the 7-deacylation product of CPC) (Figure 7C,D). CPC and DAOC were apparently more abundant in the culture supernatants.

To quantify the amount of 7-ACA more accurately, we selected 6 strains carrying genes for both CCAA and CCAB (Figure 7E). Most importantly, 7-ACA, as well as 7-ADCA, were only detected in culture supernatants of strains expressing CCAA, but not CCAB. When we consider that the peak area values of measurements from both mycelia and supernatants are not directly comparable, since mycelial extracts were concentrated (see Material and Methods) and supernatants were not, we conclude that the majority of CPC and 7-ACA are detectable in the culture supernatants. Therefore, we conducted subsequent measurements with culture supernatants.

### 3.6. Exploration of the Optimal Settings for Active Acylases

Previous studies on CCAs have demonstrated the strong pH dependency of CCA enzyme activity. For example, increased activities were obtained with alkaline pH values between 8.5 and 9.0 [22,50,58]. In the continuing analysis, we choose a 72-h culture, showing optimal expression of the *ccaA* gene (XUL-4.6). Aliquots of the supernatants were then adjusted to pH values of 9.0 and 9.5 (as described in Materials and Methods). For the incubation process, we tested various temperatures, from 8 °C to 22 °C, to optimize reaction activity but minimize degradation of CCA substrates and products. HPLC analysis was performed with the supernatant samples to measure the active CCAA one-step conversion product 7-ACA (Figure 8A). Given identical incubation temperatures, we found larger 7-ACA peaks at pH 9.5. The best results in this experiment were obtained when we incubated the supernatants for 24 h at 18 °C or 48 h at 12 °C.

All 7-ACA levels showed upward trends at 8 °C, 12 °C, and 18 °C. However, at an incubation temperature of 22 °C, net 7-ACA production decreased after 20 h of incubation at both pH values, 9.0 and 9.5. The instability of CPC and 7-ACA was more obvious with a longer period of exposure at a higher temperature (Appendix A).

A further experiment was performed to reduce the degradation of the CCA conversion product. The supernatant of CCAA (XUL-4.6) was adjusted to pH 9.5, and 7-ACA production levels were monitored at lower temperatures and prolonged incubation times (Figure 8B). When the supernatant was incubated at 4 °C or 8 °C, the net 7-ACA yield continuously increased until 144 h. For the experiment at 12 °C, the highest yield was detected at 72 h, then net 7-ACA levels were maintained and gradually decreased until 240 h. The highest yield was measured at 8 °C, pH 9.5, and 144 h of incubation, with a peak area value of 0.30 mAU*min. However, considering the time and energy costs of a long-term incubation, 48 h of incubation at 12 °C and pH 9.5 was defined in this work as the optimal condition.

### 3.7. Comparative Investigation of Transformants under Optimal Acylase Incubation Conditions

Active acylases were only found from CCAA transformants, and the net yield of CPC and the conversion product 7-ACA were measured by HPLC analysis (Figure 9A). The highest yield of 7-ACA after 48 h of incubation at 12 °C (pH 9.5) was 13.51 mg/l (XUL-4.6), while all other transformants showed different activities (Figure 9A). The same measurement was used to evaluate the concentration of CPC (Appendix A).

Both 7-ACA and CPC contents in the culture supernatants were converted to molar concentrations, and the molar ratio between 7-ACA and CPC was presented as a 100% stacked column (Figure 9B). Considering one molecule of 7-ACA is produced from one molecule of CPC, and neglecting differences in degradation, the ratios displayed in Figure 9B indicate the conversion rate of CCAA. Eventually, the one-step conversion rate of CCAA reached 30% for at least 2 strains (XUL-4.6 and XUL-4.2).

## 4. Discussion

Here, we succeeded in generating recombinant *A. chrysogenum* strains converting cephalosporin C (CPC) in vivo into 7-ACA, the precursor molecule for synthetic cephalosporins. We achieved 7-ACA production as a one-step bioconversion using a single bacterial enzyme, cephalosporin C acylase (CCA). Using codon-adapted bacterial *cca* genes optimized for efficient expression in *A. chrysogenum,* proper processing and secretion of the enzyme enabled us to determine optimal conditions for converting CPC into 7-ACA in the culture supernatant.

In the pharmaceutical industry, the production of semi-synthetic cephalosporins is based on either 7-ACA or 7-ADCA as the core beta-lactam moiety. Several genetic engineering methods have been developed to synthesize these synthons with lower financial and environmental costs. An industrial level of yield was achieved to produce adipoyl-7-aminodeacetoxy-cephalosporanic acid (ad-7-ADCA) from the penicillin producer *Penicillium chrysogenum*. With the addition of adipic acid, expression of the heterologous *cefE* gene, encoding deacetylcephalosporin C (DAC) expandase from *Streptomyces clavuligerus* induced the synthesis of ad-7-ADCA, which can be easily converted to 7-ADCA [59,60,61].

Further research was conducted on *A. chrysogenum*, resulting in a novel production method with improved product purity. The *cefEF* gene, encoding the expandase/hydroxylase was replaced by the *cefE* gene from *S. clavuligerus*, and the resulting recombinant *A. chrysogenum* strain produced a dominant yield of the beta-lactam product DAOC. DAOC undergoes further conversion to 7-ADCA through enzymatic bioconversions mediated by D-amino acid oxidase (DAO) and glutaryl acylase (GLA) [62]. These processes are equivalent to the industrial two-step enzymatic conversions of CPC to 7-ACA [13]. However, this procedure is distinct from the in vivo one-step bioconversion described in this contribution.

### 4.1. The Heterologous Genes Encoding Cephalosporin C Acylase Are Efficiently Expressed in a Fungal Host

Previous studies have reported that several heterologous genes from eukaryotic organisms were successfully expressed in *A. chrysogenum* [31,63,64,65]. For instance, overexpression of the plasma membrane H^+^-ATPase (PMA) gene from *Saccharomyces cerevisiae* under the control of the constitutive *gpdA* promoter resulted in a 1.2 to 10-fold decrease in CPC production, demonstrating that the enhanced CPC productivity during classical strain development can be attributed to an elevated ATP content due to attenuated PMA [65]. A larger gene construct, the 1.15 kb *penDE* gene cluster of *P. chrysogenum* fused with the *pcbC* promoter, was transformed into an isopenicillin N synthase-deficient mutant. Upon addition of phenylacetic acid to the culture media, the resulting transformants triggered the production of benzylpenicillin (penicillin G) in *A. chrysogenum* [63].

Until now, only a few reports have demonstrated the biosynthesis of bacterial enzymes in *A. chrysogenum*. One successful study showed the production of the oxygen-binding heme protein (hemoglobin, 15 kDa) from the bacterium *Vitreoscilla* in *A. chrysogenum*. The corresponding fungal transformants produced a 5-fold higher CPC yield than the parental strain, presumably by accelerating the oxidation reactions within the CPC biosynthesis pathway [66]. Here, we achieved one-step genetic engineering with the CPC-producing fungus *A. chrysogenum* by introducing bacterial *cca* genes via ectopic gene integration. The codon-bias of the bacterial genes was of concern as a bottleneck for optimal gene expression. Therefore, as a prerequisite for successful expression of the heterologous genes, codon optimization was taken into consideration when synthesizing the bacterial genes.

### 4.2. Bacterial Cephalosporin C Acylase Is Efficiently Processed in the Fungal Cell

To obtain the active CCA enzyme in *A. chrysogenum*, the autocatalytic cleavage process is crucial after the protein is synthesized. We confirmed the efficient cleavage of CCAA and CCAB by multiple analyses, including Western blotting, protein mass spectrometry, and enzyme activity measurements by assessing the presence of substrates and products with HPLC. These data demonstrate that *A. chrysogenum* can execute the maturation process of a bacterial enzyme. Previously, a consistent finding was reported for *A. chrysogenum* with a heterologous fungal enzyme. The expression of the gene for alkaline protease (Alp) under the control of its native promoter from *Fusarium* sp. S-19-5 resulted in the secretion of the heterologous enzyme. This was feasible with the guidance of the signal peptide from *Fusarium* [64].

An unexpected finding arose from the protein mass spectrometry analyses. In addition to the cleavage between the α- and β-subunit, an extra truncation was observed for CCAA at the N-terminus of the α-subunit, resulting in loss of the Hisx6-tag, which explains why we failed to detect the α-subunit in Western blots. Since Ser1β, the N-terminal residue of the β-subunit is known to orchestrate the intramolecular cleavage of the enzyme precursor, we speculate that Ser4α is responsible for the unexpected His-tag truncation. Mimicking the proteolytic mechanism of Ser1β, Ser4α may act as the nucleophile and proton donor during the autoproteolytic cleavage. An alternative possibility is that the α-subunit was degraded by some intrinsic fungal protease [67]. However, the cleaved-off peptide was rather short and not located at the core catalytic positions or the substrate contacting areas. Thus, the impact of the CCAA truncation on enzyme activity is probably negligible.

Full processing of another acylase was previously reported by Honda et al. [68]. There, both the α- and β-subunits of the wild-type acylase II from *Pseudomonas* sp. SE83 were detected in Western blot analysis using anti-acylase II antiserum. These authors found a correlation between acylase activity and genomically integrated acylase genes; however, they did not measure the in vivo conversion of CPC into 7-ACA.

In the entire study, we did not detect the cleavage process of CCAC in any of the tested transformants. Possibly, the inhibiting factor for correct intracellular cleavage of CCAC was the redundant bacterial signal peptide at the N-terminus of the CCAC precursor. Therefore, we introduced a deletion mutation in the *ccaC* expression vector to eliminate the non-functional signal peptide. However, the CCAC precursors from the resulting mutant transformants were not processed either (Appendix A), and hence further investigations are needed into the cleavage mechanism.

### 4.3. Conversion of CPC into 7-ACA in the Culture Supernatant Has Applied Relevance

An increasing number of fungal species have been adopted for microbial biosynthesis in the pharmaceutical and food industry, thus replacing traditional chemical processes. Except for the widely known producer *Saccharomyces cerevisiae,* a representative example is the riboflavin (Vitamin B2) and folic acid (Vitamin B9) producer *Ashbya gossypii* [69,70,71,72]. Through a series of metabolic engineering and mutant selection processes, the productivity has been increased dramatically to grams per liter scales [69,70]. A genetically engineered *A. gossypii* strain achieved complete secretion of riboflavin into the extracellular medium via specific carriers [72].

In this study, we detected an active cephalosporin C acylase in the culture supernatants, although no functional fungal signal peptide was identified in the bacterial enzyme. In addition, CPC produced in the fungal cytoplasmic space is immediately transferred to the extracellular space, providing the acylase easy access to its enzymatic substrate. These findings imply that enhancing secretion of the bacterial enzyme to the culture medium might be an effective route to improving straightforward 7-ACA production.

### 4.4. The Heterologous Cephalosporin C Acylase Shows Enzymatic Activity

After *cca* gene expression and protein maturation, the processed acylase precursor forms the α and β heterodimer structure, enabling direct conversion of CPC to 7-ACA. Microbial production of 7-ACA by combining fungal and bacterial genes has been reported in earlier research [73]. The D-amino acid oxidase (DAO) gene from *Fusarium solani* and the cephalosporin C acylase gene from *Pseudomonas diminuta* were expressed under the control of regulatory elements from the alkaline protease gene from *A. chrysogenum*, encoding a secretory enzyme. However, the resulting transformants synthesized and secreted commercially insignificant levels of 7-ACA.

Using optimal fermentation and incubation conditions, we were able to reach an in vivo one-step enzymatic conversion rate of CPC to 7-ACA of up to 30%. Another report previously showed a similar conversion rate of CPC to 7-ACA in *A. chrysogenum* [74]. However, this study differs in a number of ways from our work. Firstly, Liu et al. used a mutant variant of another class III CCA from *Pseudomonas N176* and synthesized a codon-optimized gene, which was expressed in both *E. coli* and *A. chrysogenum*. Southern blot analysis detected single copies of the foreign gene in two transformants. Here, we detected multiple integrations of the heterologous gene, similar to previous comparable studies in *A. chrysogenum* [29,68]. Secondly, Western blot analysis of the two fungal transformants showed no α-subunit, which, however, was detectable in transgenic *E. coli* strains [74]. In our analysis, we used extensive MS analysis to clearly demonstrate the presence of the α-subunit, albeit truncated at the N-terminus with the loss of three amino acids. Thirdly, Liu et al. claimed a 30% conversion of CPC into 7-ACA. However, they do not indicate whether they used samples from mycelial extracts or supernatants. Using more than 18 fungal transformants, we demonstrated the expression of the heterologous genes by Western blots. Finally, we found only minor amounts of CPC in the mycelial extracts, whereas most of the antibiotics reside in the supernatants. Therefore, we measured the conversion of CPC into 7-ACA in culture media supernatants under different physical conditions.

Despite the successful cleavage process of CCAB in *A. chrysogenum* mycelia and supernatants, 7-ACA was not detected after incubation. Notably, the CCAB protein amount (XUL-22.3) in the culture supernatant was much lower than that of CCAA (XUL-4.6). Moreover, considering that a rather high amount of CPC remained in the reaction pool, the rate-limiting factor is not depletion of the substrate but rather the limited amount of enzyme. Certainly, an adequate amount of active enzyme is a prerequisite for obtaining detectable levels of 7-ACA in the microbial production process. As a slow-growing fungus, the synthesis of the heterologous enzyme in *A. chrysogenum* is much slower than in the bacterial producer *E. coli,* and thus exhibits comparatively lower productivity.

For future strategies to improve the whole production process, we suggest fusion constructs, where the acylase gene is in frame with a fungal gene encoding an efficiently secreted endogenous protein. Other strategies are the increase of the gene copy number or the introduction of a strong inducible promoter. Finally, the use of protease deficient or high-protein-secreting recipient strains will be further options for increasing the yield of 7-ACA. We did not aim in our study to reach an industrial level of productivity. It can be foreseen that the transfer of our approach to fermentation processes in large stirred tank reactors will certainly require the adaptation of the whole culture conditions, including the optimization of growth media.

Taken together, the significant value of our work is that all the sophisticated downstream procedures were avoided by expressing the heterologous cephalosporin C acylase in the CPC-producing fungus. This dramatically reduces manufacturing costs and energy consumption. We consider our method to be a valuable technological contribution to streamlining future pharmaceutical manufacturing processes.

## Figures and Tables

**Figure 1 jof-08-00450-f001:**
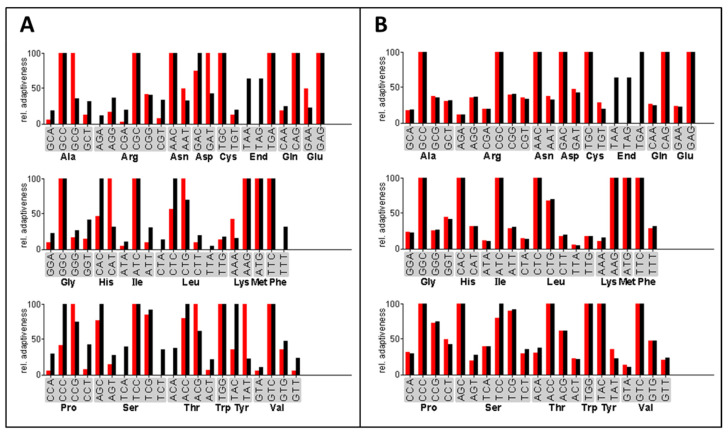
Example of the codon-optimizing process. Shown is the codon usage between the *ccaC* gene and the *A. chrysogenum* genome before (**A**) and after (**B**) codon optimization. The red bars represent the relative codon usage of the *ccaC* gene from *Pseudomonas* sp. GK16, the black bars indicate that of the *A. chrysogenum* genome database. The relative adaptiveness values of the various codons coding the same amino acid were normalized such that the most abundant one is 100%.

**Figure 2 jof-08-00450-f002:**
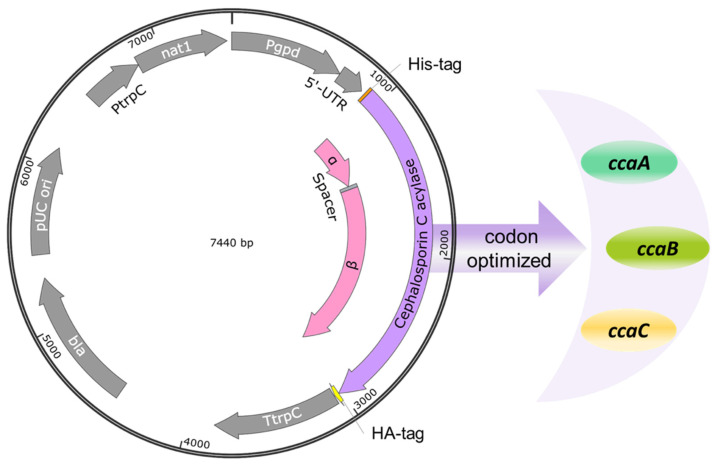
Fungal expression vectors (Table 2) for the recombinant CCA production in *A. chrysogenum*. Three codon-optimized *cca* genes (*ccaA*, *ccaB*, *ccaC*) were fused 5′ with a Hisx6-tag sequence and 3′with an HA-tag. The *cca* genes were connected with the tag sequences by GGCGGTGGTGGCTCA-linkers encoding Gly-Gly-Gly-Gly-Ser. The expression of the recombinant proteins was under the control of the constitutive *gpdA* promoter. The nourseothricin resistance gene served as a selectable marker in fungal transformations.

**Figure 3 jof-08-00450-f003:**
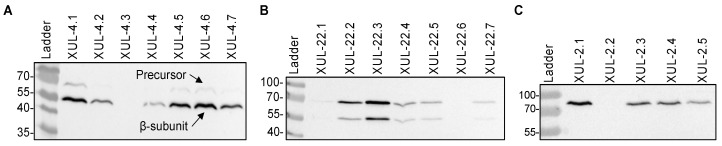
Western blot analysis to detect recombinant CCAs (CCAA, CCAB, and CCAC) fused to the HA-tag in *A. chrysogenum* transformants as indicated. (**A**) Detection of CCAA in mycelial extracts. The protein bands of CCAA precursor (79 kDa) and β-subunit (60 kDa) are marked. (**B**) Detection of CCAB in mycelial extracts. (**C**) Detection of CCAC precursor in mycelial extracts.

**Figure 4 jof-08-00450-f004:**
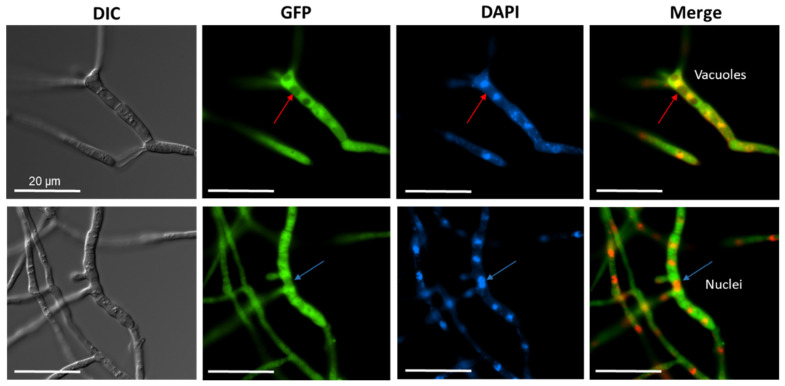
Fluorescence microscopy to demonstrate the cytoplasmic localization of CCAA. The *ccaA* gene was fused to the *gfp* gene (EGFP) at the 3′ terminus. DAPI staining was performed for the visualization of nuclei and mitochondria. Red arrows indicate putative vacuoles, blue arrows indicate nuclei. Merged images with EGFP filter and DAPI filter indicate the cytoplasmic localization of CCAA. All scale bars correspond to 20 µm.

**Figure 5 jof-08-00450-f005:**
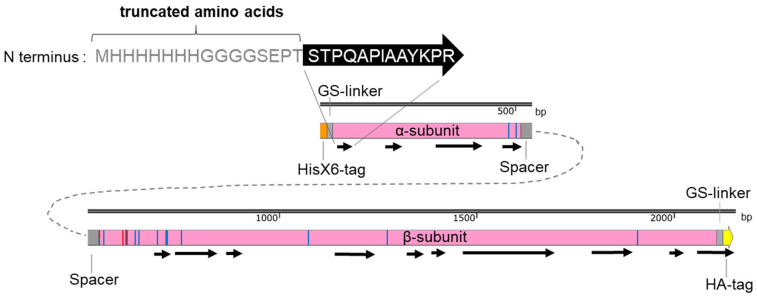
Protein mass spectrometry analysis of trypsin digested total fungal proteins for verifying CCA expression in *A. chrysogenum*. The total protein extract of fungal mycelia was separated using SDS-PAGE gel, and the gel slices containing proteins of specific size were analyzed by protein MS. Black arrows indicate the identified peptides. Amino acids at the truncated N-terminal region of the α-subunit are given. The catalytic amino acids are indicated as red bars, and the amino acids, which have contact with the CCA substrate, are marked by blue bars.

**Figure 6 jof-08-00450-f006:**
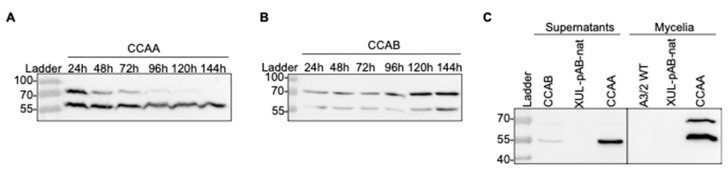
Extensive investigation of the CCA synthesis and processing in *A. chrysogenum.* (**A**) The expression and cleavage process of CCAA over time from mycelia of transformant XUL-4.6. (**B**) Time course of expression and cleavage of CCAB from transformant XUL-22.3. (**C**) Detection of processed CCAA (XUL-4.6) and CCAB (XUL-22.3) in extracts from mycelia and supernatants. As indicated, samples from the wild-type (A2/3 WT) or transformants carrying pAB-nat plasmid (XUL-pAB-nat) served as controls.

**Figure 7 jof-08-00450-f007:**
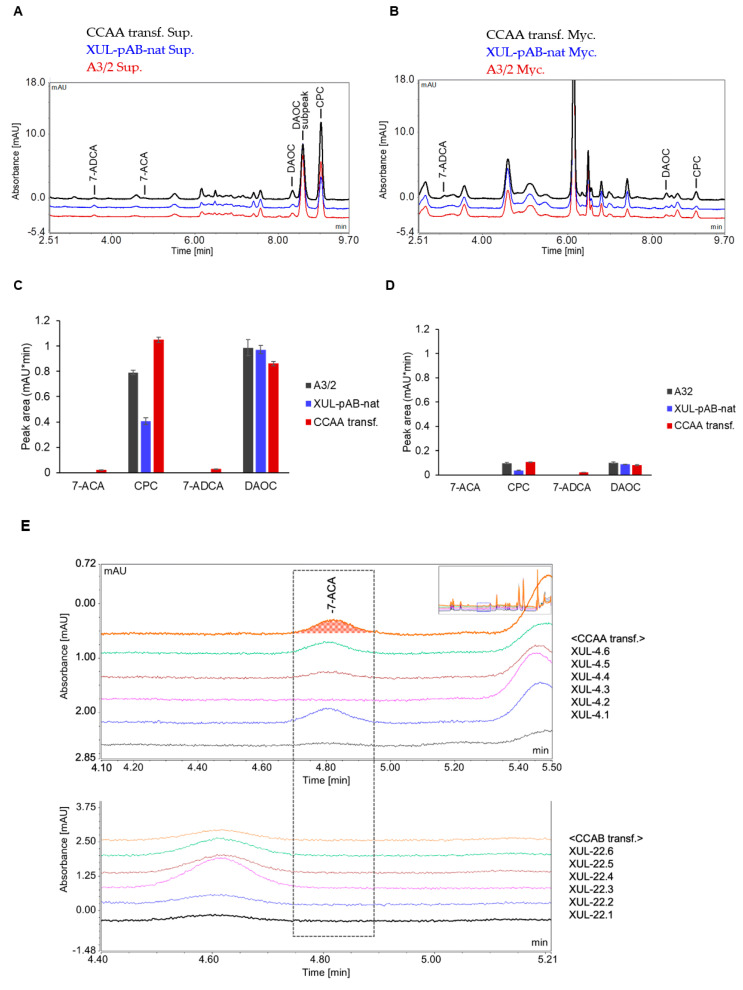
HPLC analysis of fungal mycelia and culture supernatants to separately detect compounds related to the CPC biosynthesis pathway. (**A**) HPLC spectrum of fungal culture supernatants. (**B**) HPLC spectrum of concentrated mycelial extracts prepared as described in the Materials and Methods section. (**C**,**D**) HPLC peak areas of selected substances detected in the supernatants (**C**) and mycelial samples (**D**). (**E**) Detection of 7-ACA in the culture supernatants of randomly selected CCAA and CCAB transformants as indicated. For HPLC measurements, fungal mycelia and culture supernatants from 6 days cultures were separated by vacuum filtration. The sample preparation process was followed as described in the Material and Methods section.

**Figure 8 jof-08-00450-f008:**
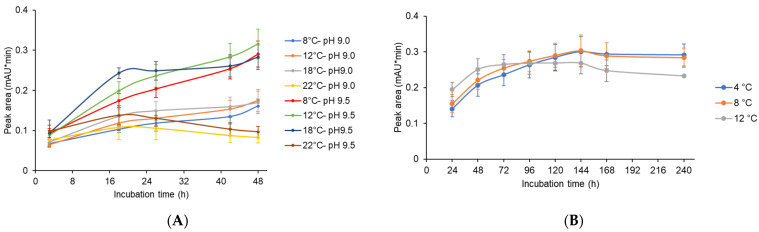
Post-fermentation approaches to obtain significant productivity of CCAA (XUL-4.6). (**A**) Optimization of temperature and pH conditions for the deacylation reaction of CCA. The supernatants of 72 h of culture were modified to pH 9.0 and pH 9.5, which are close to the preferred environment for the deacylation, and incubated at different temperatures from 8 °C to 22 °C for up to 48 h. Samples were taken at different time points and HPLC analysis was followed, and the peak areas of CCAA deacylation product 7-ACA were plotted. For every condition, at least three independent experiments were conducted. (**B**) The maximal production of 7-ACA was observed for the extended incubations at low temperatures. The supernatants were adjusted to pH 9.5 and incubated at 4 °C to 12 °C for up to 240 h.

**Figure 9 jof-08-00450-f009:**
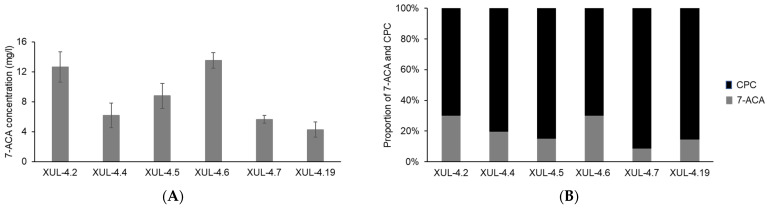
7-ACA yield and the conversion rate of CCAA transformants after incubating at optimal reaction conditions. The supernatants of CCAA transformants were isolated after 72 h of culture and the pH was adjusted to 9.5. HPLC analyses were performed with samples collected after 48 h of incubation at 12 °C. (**A**) The concentration of CCA product 7-ACA was calculated according to the calibration curve (Appendix A). (**B**) The proportion of the concentration of 7-ACA and CPC after enzyme reaction represents the conversion rate of CCA. The concentration of CPC was calculated using the reference calibration curve (Appendix A).

**Table 1 jof-08-00450-t001:** *Acremonium**chrysogenum* strains used in this work.

Strains	Genotypes	Source
A3/2	Producer strain, *nat^s^*	[29]
XUL-4.1, -4.2, -4.3, -4.4, -4.5, -4.6, -4.7, -4.19,	A3/2, pXUL-4 (*ccaA*), *nat^r^*	This work
XUL-22.1, -22.2, -22.3, -22.4, -22.5, -22.6, -22.7	A3/2, pXUL-22 (*ccaB*), *nat^r^*	This work
XUL-2.1, -2.2, -2.3, -2.4, -2.5,	A3/2, pXUL-2 (*ccaC*), *nat^r^*	This work
XUL-pAB-nat	A3/2, pAB-nat, *nat^r^*	This work

*nat^s^*, nourseotricin susceptible; *nat^r^*, nourseotricin resistant.

**Table 2 jof-08-00450-t002:** Plasmids used in this work.

Plasmids	Genotypes	Source
pEX K248 >CCAA	HA::*ccaA*::HA, *kan^r^, amp^s^*	This work ^(1)^
pEX-K248->CCAB	His::*ccaB*-HA, *kan^r^, amp^s^*	This work ^(1)^
pEX-K168->CCAC	HA::*ccaC*::HA, *kan^r^, amp^s^*	This work ^(1)^
pGG-C-EGFP	P*gpdA*::*egfp*::T*trpC*, *nat^r^, kan^s^, amp^r^*	[35]
pAB-nat	P*gpdA*::T*trpC*, *nat^r^, kan^s^, amp^r^*	Modified from pGG-C-EGFP [36]
pXUL-2	P*gpdA*::HA::*ccaC*::HA::T*trpC*, *nat^r^, kan^s^, amp^r^*	This work
pXUL-4	P*gpdA*::His::*ccaA*-HA::T*trpC*, *nat^r^, kan^s^, amp^r^*	This work
pXUL-10	P*gpdA*::His::*ccaA*::HA::*egfp*::T*trpC*, *nat^r^, kan^s^, amp^r^*	This work
pXUL-22	P*gpdA*::His::*ccaB*::HA::T*trpC*, *nat^r^, kan^s^, amp^r^*	This work

^(^^1)^ The corresponding genes were synthesized with codon usage adapted to the *A. chrysogenum* genome.

**Table 3 jof-08-00450-t003:** Information about CCAs studied in this work ^(^^1)^.

Name	Class	Source Strain	Predicted Molecular Weight (kDa)	Subunit Structure α + β (kDa)	Spacer (aa)	References
CCAA	I	*Pseudomonas* sp. GK16	79	19 + 60	10	[51]
CCAB	I	*Pseudomonas* sp. GK16	79	19 + 60	10	Patent 2014, CN103937764B, (Amicogen Inc., Jinju, South Korea)
CCAC	III	*Pseudomonas* sp. SE 83 (AcyII)	84	24 + 60	10	Patent 2009, US7592168B2, (Sandoz AG, Basel, Switzerland)

^(1)^ The corresponding genes were synthesized with codon usage adapted to the *A. chrysogenum* genome.

## Data Availability

Not applicable.

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
