# Peer review of "A Straightforward Approach to Synthesize 7-Aminocephalosporanic Acid In Vivo in the Cephalosporin C Producer Acremonium chrysogenum"

_jof, 2022, doi:10.3390/jof8050450_

Round 1

Reviewer 1 Report

This is a well constructed paper. In the abstract on the 3rd line from the bottom, should the word be strain rather than stain?

Have you patented this methodology? What are the plans for the future with this method? It would be good to see what are the next plans in the conclusion? Just a brief one or 2 lines. It is important that this research is progressed to the next level based on the results of this study.  

Author Response

Review 1:

This is a well constructed paper. In the abstract on the 3rd line from the bottom, should the word be strain rather than stain? Was done

Thank you for the very positive response.

Have you patented this methodology? No, the method was not yet patented.

What are the plans for the future with this method? It would be good to see what are the next plans in the conclusion? Just a brief one or 2 lines. It is important that this research is progressed to the next level based on the results of this study.  

We have added text on future plans and these can be found in the discussion at line 623-631

For future strategies to improve the whole production process we suggest fusion constructs, where the acylase gene is in frame with a fungal gene encoding an efficiently secreted endogenous protein. Other strategies are the increase of the gene copy number or the introduction of a strong inducible promoter. Finally, the use of protease deficient or high-protein-secreting recipient strains will be further options for increasing the yield of 7-ACA. We did not aim in our study to reach an industrial level of productivity. It can be foreseen that the transfer of our approach to fermentation processes in large stirred tank reactors will certainly require the adaptation of the whole culture conditions, including the optimization of growth media.

Reviewer 2 Report

This research study was well presented and worthwhile.

Author Response

Review 2:

This research study was well presented and worthwhile.

Thank you for the very positive response.

Reviewer 3 Report

the authors report the possibility of using a new approach to synthesize 7-aminocephalosporic acid in vivo in Acremonium chrysogenum

The experimental design is well constructed and described. The results are well represented.

Some considerations:

lines 252 and 253. sp should not be written in italics

Figure 8. In the comparison of time, temperature and pH, the authors report 12 ° C at 48h as the best. However, it is not clear from the graph if they are significant differences. 40 h could be more advantageous.

I recommend adding the statistic.

Author Response

Review 3:

the authors report the possibility of using a new approach to synthesize 7-aminocephalosporic acid in vivo in Acremonium chrysogenum

The experimental design is well constructed and described. The results are well represented.

Some considerations:

lines 252 and 253. sp should not be written in italics Was done as requested throughout the text

Figure 8. In the comparison of time, temperature and pH, the authors report 12 ° C at 48h as the best. However, it is not clear from the graph if they are significant differences. 40 h could be more advantageous.

Thank you for your carefully made comment. We looked again over Fig. 8.A and found that the “orange” and the “green”, indicating our measurement at 12°C, are higher after 48h than after 40 or 42 hours. We suggest that the different colours chosen in our graphs are a little bit confusing.

I recommend adding the statistic.

Line 448 (legend of figure): “For every condition, at least three independent experiments were conducted.”

Reviewer 4 Report

The authors present a paper that presents a the production of cephalosporin C acylase (CCA) with genetic sequences which were introduced by vectors into the fungi Acremonium chrysogenum.

the author use genetic techniques, chromatography, optimized temperature and pH and other techniques to determine the conditions which give the best CCA.

Manuscript well written and methods well presented the conclusions are in line with work done

Author Response

Review 4:

The authors present a paper that presents a the production of cephalosporin C acylase (CCA) with genetic sequences which were introduced by vectors into the fungi Acremonium chrysogenum.

the author use genetic techniques, chromatography, optimized temperature and pH and other techniques to determine the conditions which give the best CCA.

Manuscript well written and methods well presented the conclusions are in line with work done

We thank the reviewer for the very positive response regarding your work.